# Notch Signaling and PD-1/PD-L1 Interaction in Hepatocellular Carcinoma: Potentialities of Combined Therapies

**DOI:** 10.3390/biom14121581

**Published:** 2024-12-11

**Authors:** Annapaola Montagner, Andrea Arleo, Fabrizia Suzzi, Antonino B. D’Assoro, Fabio Piscaglia, Laura Gramantieri, Catia Giovannini

**Affiliations:** 1Department of Medical and Surgical Sciences, Bologna University, 40138 Bologna, Italy; andrea.arleo4@unibo.it (A.A.); fabrizia.suzzi3@unibo.it (F.S.); fabio.piscaglia@unibo.it (F.P.); catia.giovannini4@unibo.it (C.G.); 2Department of Oncology, Mayo Clinic College of Medicine, Rochester, MN 55902, USA; dassoro.antonio@mayo.edu; 3Division of Internal Medicine, Hepatobiliary and Immunoallergic Diseases, IRCCS Azienda Ospedaliero-Universitaria di Bologna, 40138 Bologna, Italy; laura.gramantieri@aosp.bo.it

**Keywords:** Notch, PD-1, PD-L1, hepatocellular carcinoma (HCC), immunotherapy, combination therapy

## Abstract

Immunotherapy has shown significant improvement in the survival of patients with hepatocellular carcinoma (HCC) compared to TKIs as first-line treatment. Unfortunately, approximately 30% of HCC exhibits intrinsic resistance to ICIs, making new therapeutic combinations urgently needed. The dysregulation of the Notch signaling pathway observed in HCC can affect immune cell response, reducing the efficacy of cancer immunotherapy. Here, we provide an overview of how Notch signaling regulates immune responses and present the therapeutic rationale for combining Notch signaling inhibition with ICIs to improve HCC treatment. Moreover, we propose using exosomes as non-invasive tools to assess Notch signaling activation in hepatic cancer cells, enabling accurate stratification of patients who can benefit from combined strategies.

## 1. Introduction

Liver cancer is the sixth most frequently diagnosed malignancy and the third cause of cancer-related deaths worldwide. It is a highly fatal tumor with a global mortality-to-incidence ratio of 0.88 [1]. Hepatocellular carcinoma (HCC) represents the most common primary liver cancer, accounting for 75–85% of cases, followed by intrahepatic cholangiocarcinoma (10–15% of cases) [1].

In most patients, HCC develops in a context of chronic liver diseases, with hepatitis B (HBV), hepatitis C (HCV), alcohol-related liver disease (ARLD), and metabolic-dysfunction-associated steatotic liver disease (MASLD) being the primary risk factors. Dietary exposure to aflatoxin B1 constitutes another key determinant in HCC occurrence [2].

Early- and intermediate-stage HCC can be treated by locoregional therapies, hepatic resection, or liver transplantation. However, disease progression to advanced stages, following repeated relapses (recurrence rate of 70% at 5 years), is frequently accompanied by cases of late diagnosis due to the initially asymptomatic clinical course of HCC [3]. The management of advanced unresectable HCC involves pharmacological systemic therapies that rely on tyrosine kinase inhibitors, immune checkpoint inhibitors (ICIs), or anti-VEGF drugs. The current standard of care for first-line treatment is an ICI-based combination immunotherapy employing atezolizumab (anti-PD-L1 antibody) and bevacizumab (anti-VEGF antibody) [4]. Recently, another ICI-based therapeutic combination, tremelimumab (anti-CTLA-4 antibody) plus durvalumab (anti-PD-L1 antibody), has been approved as first-line treatment for unresectable HCC [5].

The available therapies show a response rate of approximately 20–40%, due to primary resistance, while some patients who initially respond eventually develop acquired resistance [6]. Although ICI-based treatments have significantly improved the prognosis of advanced HCC, their effectiveness is influenced by the tumor microenvironment’s (TME) complexity and immunological heterogeneity [7]. The TME is indeed an intricate cluster of different cell populations influencing the amount and functionality of neighboring cells. Every cell population found in the TME, including stromal and immune cells, affects the response to ICIs. In addition, tertiary lymphoid structures—comprising B cells, T cells, and dendritic cells—have recently emerged as key factors associated with ICI responsiveness [8]. The interaction between cancer cells and the extracellular environment orchestrates tumor development and can lead to treatment failure.

An accurate understanding of the crosstalk between HCC cancer cells and the TME is crucial for identifying more effective therapeutic strategies. In this regard, we will focus on the role of aberrant Notch signaling in liver cancer and its impact on the immune cells composing the TME.

The Notch pathway is one of the major developmental signals that control essential cellular processes, including proliferation, differentiation, and migration during embryogenesis. In addition, it preserves the integrity of adult tissues by regulating stem cell populations within their niches. Due to its pivotal role, dysregulation of the Notch pathway can contribute to the onset of different human diseases, including cancer [9]. The Notch signaling pathway is known to be mainly affected by interactions with other stem-cell-related pathways, such as Hedgehog, Wnt, and TGF-β. However, recent studies suggest an emerging interplay between Notch signaling and the PD-1/PD-L1 axis. Here, we will highlight the evidence of communication between Notch signaling and the PD-1/PD-L1 axis, and the inter-regulatory dynamics that modulate their signals. Gaining insights on their interconnection could provide solid bases for incorporating Notch inhibitors into combination therapies with ICIs, to overcome drug resistance and enhance the efficacy of HCC treatments.

## 2. Notch Signaling Pathway Overview

The Notch pathway is an evolutionarily conserved juxtracrine signaling system comprising three major components: four type I transmembrane Notch receptors (Notch1, 2, 3, and 4), five ligands (Jagged 1 and 2 and Delta-like 1, 3, and 4), which are also expressed as membrane-bound proteins, and downstream effectors of Notch signaling [10].

Notch receptors are presented on the cell surface as non-covalently associated heterodimers originally synthesized as single precursors, then processed in the Golgi apparatus by a Furin-like convertase into two segments (S1 cleavage), and subsequently transported as mature receptors to the cell membrane [11]. The extracellular domain (ECD) of the Notch receptor harbors 29 to 36 epidermal growth factor (EGF) repeats, three Lin12-Notch Repeats (LNRs), and a heterodimerization domain (HD) located adjacent to the cell membrane. The transmembrane polypeptide (TMF) encompasses a small extracellular portion, a membrane-spanning region, and the Notch intracellular domain (NICD). The NICD contains several functional regions, including a RAM domain, seven ankyrin repeats (ANKs) responsible for interaction with nuclear transcription factors, two nuclear localization signals (NLSs), a transcription activation domain (TAD) found in Notch1 and 2, but not in 3 and 4, and a PEST motif, essential for NICD proteolytic degradation [11,12,13].

The structure of Notch ligands is similar to the structure of Notch receptors, featuring many EGF-like repeats (6 and 8 in Dlls and 18 in Jagged ligands), an amino-terminal DSL (Delta, Serrate, and Lag2) domain, which contains the Notch-binding site, and a cysteine-rich domain (CR) in jagged ligands located near the transmembrane subunit [14] (Figure 1a).

Canonical pathway activation occurs upon ligand–receptor binding between neighboring cells and exerts a regulatory effect on both signal-sending and signal-receiving cells [15]. The N-terminal DSL domain of the ligand interacts with the EGF repeats of the Notch receptor’s extracellular domain, triggering a conformational change in the protein and exposing a metalloprotease target site within the LNR region [12]. A first proteolytic cleavage is performed by the disintegrin and metalloprotease ADAM/TACE (S2 cleavage), releasing an extracellular fragment of the receptor still bound to the ligand, which is internalized by the neighboring ligand-expressing cell. Simultaneously, the residual Notch c-terminal fragment, called Notch extracellular truncation (NEXT), is endocytosed into the signal-receiving cell. Studies suggest that ligand binding alone is insufficient for pathway activation, which is instead the result of the pulling force exerted by the signal-sending cells initiating complex endocytosis upon ligand engagement and inducing the receptor’s conformational change [13]. The second sequential proteolytic cleavage (S3 cleavage) occurs within the transmembrane domain of NEXT in the endosome and is catalyzed by a presenilins multi-protein complex with y-secretase activity. This generates the active NICD, which is released into the cytoplasm and translocates into the nucleus to form a protein complex with the transcription factor CSL (also known as RBP-J/CBF1), promoting the expression of Notch target genes. In the absence of NICD, CSL acts as a transcriptional repressor by recruiting co-repressor proteins, such as SMRT/N-CoR and histone deacetylases (HDACs). NICD-CSL interaction prompts the displacement of the co-repressors and the simultaneous engagement of activating partners, with Mastermind-like protein (MAML) serving as a core co-activator. The NICD-CSL-MAML ternary complex recruits other transcriptional activators, like histone acetyltransferase (HATs), thus creating an activator complex that fosters gene transcription [12,14] (Figure 1b).

Non-canonical signaling can be activated without ligand–receptor interaction, and the Notch signal transduction may occur through NICD in a CSL-independent route, or without NICD release but in the presence of CSL. In a third type of non-canonical Notch pathway, neither NICD nor CSL is involved [16].

Major canonical targets of Notch signaling are members of the HES (hairy/enhancer of split), HEY (HES-related repressor protein), and HERP gene families encoding for helix–loop–helix transcriptional repressors. Conversely, regulation of other targets has also been observed, including cell cycle proteins such as Cyclin D1, Cyclin A, and c-Myc, members of the NF-κB family, p21, SNAI, ubiquitin ligase SKP2, PI3K/AKT, and insulin-like growth factor 1 (IGF-1) and its receptor (IGF-1R). This regulation may be the result of intense crosstalk with other signaling pathways [15,17,18]. The list of genes and proteins directly or indirectly regulated by Notch can greatly vary and be context-specific, reflecting the considerably diverse biological effects induced by Notch signaling. The variation in the downstream signaling output is also generated by fine pathway regulation via post-translational modifications [19]. Notch receptors are subjected to ubiquitinylation mediated by E3 ubiquitin ligases, which regulates their recycling and degradation, thus influencing their accessibility to ligands [13]. Likewise, the amount and localization of Notch ligands on the cell surface are critical for signal activation and are regulated by a complex system involving ubiquitinylation and epsin-mediated endocytosis [20]. The strength of Notch signaling is also modulated through NICD proteasomal degradation following PEST motif phosphorylation and protein ubiquitinylation [19].

The peculiarity of this cell–cell communication pathway is that the signal is directly transduced from the membrane to the nucleus through an activated fragment of the receptor, in the absence of second messengers for signal amplification. This mechanism provides cells with a high signal sensitivity and allows precise tuning of signal strength [19,21]. Thus, hyper- or hypo-activation of Notch signaling can lead to phenotypes associated with developmental defects and many pathological conditions [21].

## 3. Notch Signaling in Epithelial Cells

The high level of conservation of the Notch pathway in metazoans unveils its crucial role in development. Notch signaling is active during embryonic and postnatal development, controlling cell fate and tissue boundaries through the mechanisms of lateral inhibition, lateral induction, and cell lineage decision [20]. In adult tissues, it is involved in the renewal and differentiation of stem cells and progenitor cells, ensuring physiological tissue homeostasis. Upon organ injury or stress, the Notch pathway is activated to orchestrate regeneration and damage repair [12,19].

Several studies have emphasized the crucial role of Notch signaling in the maintenance of stem cells in the gastrointestinal tract. Notch1 and Notch2 have been recognized as predominant regulators of intestinal stem cell homeostasis, as well as gastric stem cell proliferation, differentiation, and apoptosis [22,23]. In addition, the Notch pathway is involved in the differentiation of breast epithelial cells during mammary gland development [17]. Notch1 and DLL1 have been reported to control the self-renewal ability of mammary stem cells, while Notch1, -2, and -3 are involved in the differentiation of bipotent progenitor cells into the luminal cell lineage [15,24]. The pivotal role of Notch signaling in prostate development is demonstrated by the dynamic changes of Notch1 expression during prostate growth and differentiation. Shou et al. detected high levels of Notch1 mRNA expression in early postnatal epithelial cells during rodent prostatic development, with downregulation in the mature organ [25]. Another study by the same authors demonstrated that Notch signaling is essential for the maintenance and re-growth of the adult prostate [26].

Given the critical roles played by Notch signaling in the renewal and maintenance of multiple tissues and organs, dysregulation of the Notch pathway has increasingly been associated with tumor onset and progression [27]. Aberrant expression of Notch receptors, ligands, and targets has been observed in several epithelial forms of cancer, including but not limited to gastric, lung, liver, pancreatic, head and neck, skin, breast, and prostate carcinomas, with Notch acting as either an oncogene or tumor suppressor depending on the cellular context and type of malignancy [28,29].

Aberrant Notch signaling can be caused by mutations in Notch genes. Approximately 5–10% of triple-negative breast cancer (TNBC) tumors feature activity-enhancing mutations in the heterodimerization domain (DM) or in the PEST motif of Notch1 and Notch2 receptors [30]. Likewise, a fraction (around 10%) of non-small-cell lung carcinomas (NSCLCs) exhibit gain-of-function mutations in the Notch1 gene [31]. In small-cell lung cancer (SCLC), the prevalence of loss-of-function mutations of Notch proteins is reported to be about 25–28%, with Notch1 being the most frequently mutated, followed by Notch2, Notch4, and Notch3 [32]. Activity-reducing mutations in Notch1, -2, and -3 are also found in skin (cSCC), head and neck (HNSCC), esophagus (ESCC), and lung squamous cell carcinomas [28].

Dysregulation of Notch signaling in the absence of genetic mutations in the pathway is even more frequently observed; however, the causes of these alterations remain poorly understood.

The Notch pathway can be activated by stress factors, such as hypoxia, a common feature of solid malignancies, and may contribute to hypoxia-induced epithelial-to-mesenchymal transition (EMT), a key mechanism involved in cancer progression [28]. A recent study has shown that hypoxia increases the expression of Delta-like 4 and Notch4 in lung adenocarcinoma cells, promoting aberrant proliferation, migration, and inhibition of cell apoptosis via the ERK/JNK/P38 MAPK pathway [33].

Several other studies identify Notch as a crucial regulator of EMT and the seeding of organ metastases [34,35]. Notably, Notch3 expression has been linked to EMT-induced cancer cell plasticity and the emergence of lung metastases in breast cancer models [34]. Similarly, in prostate cancer, significantly increased expression of Jagged1 and Notch1 in metastatic tumors, compared to primary lesions, supports the involvement of the Jagged1-Notch1 axis in progression through the induction of the EMT phenotype [36]. Likewise, in head and neck squamous carcinoma (HNSCC), upregulation of the Notch4-HEY1 pathway is associated with EMT [37]. However, in esophageal squamous cell carcinoma, Notch3 negatively regulates EMT. Matsuura et al. demonstrated that receptor downregulation results in the activation of EMT and the development of chemotherapy resistance in ESCC cells [38].

Overall, these findings highlight the considerable divergent outcomes produced by the Notch pathway in a time-, gene-, and cell-type-dependent manner [19]. Indeed, different Notch receptors could play opposite roles in tumorigenesis and act as oncogenes or tumor suppressors, even in a stage-dependent manner [28,39] (Table 1). This pleiotropy might result from epigenetic transcriptional modulation and Notch signaling crosstalk with other key oncogenic pathways, both of which are context-dependent [21].

Moreover, emerging evidence highlights a connection between the Notch pathway and resistance to anticancer therapy. Notch signaling promotes resistance by enriching and sustaining a small subset of cancer cells with enhanced stem-like features (CSCs), characterized by higher expression of multidrug resistance genes and anti-apoptotic proteins, and fostering the transition towards the aggressive mesenchymal-like phenotypes [41,42]. For instance, in pancreatic cancer, it has been demonstrated that gemcitabine-resistant cells acquire the EMT phenotype and exhibit a high upregulation of Notch2 and Jagged1 [40]. Furthermore, Below and Osipo reported that inhibition of the Notch pathway can prevent or reverse resistance by reducing the stem-like population of breast cancer cells (BCSCs) [41].

## 4. Notch Signaling in the Liver

As described for other tissues and organs, many studies have established that Notch signaling plays a fundamental role in liver embryogenesis, determining the differentiation of hepatoblasts into the biliary or hepatocyte lineage, and controlling the correct morphogenesis of the biliary tree. The involvement of Notch signaling in bile duct development, especially of the Notch2 receptor and Jagged1 ligand, was demonstrated by Alagille Syndrome (ALGS). This disorder, characterized by a paucity of intrahepatic bile ducts as a hallmark, is associated with mutations in Notch2 and Jagged1 [19]. Flynn et al. have also suggested a crucial role for Notch3 in bile duct formation [43].

Additionally, the Notch pathway is activated in liver disease to coordinate repair mechanisms aimed at restoring the architecture and function of the hepatic lobules and biliary tree. Notch signaling, together with other morphogenic pathways such as Wnt/β-catenin and Hedgehog, guides injury-induced liver repair by regulating various types of hepatic cells. Particularly noteworthy is the Notch-mediated activation and expansion of adult hepatic progenitor cells (HPCs), which are called into action in response to liver injury when the proliferative ability of hepatocytes and cholangiocytes is impaired [19,44].

Compelling evidence supports a pivotal role for the Notch pathway in physiological angiogenesis, determining the differentiation of endothelial and vascular smooth muscle cells, and regulating the sprouting process [45]. Its involvement in the vascularization can be observed in the activity of two Notch ligands, Dll4 and Jagged1, which seem to have opposite effects on angiogenesis. Notably, Jagged1 can bind and activate Notch1 to stimulate angiogenesis by enhancing VEGF activity, whereas the interaction between Dll4 and Notch receptors restricts the sprouting of endothelial cells in growing blood vessels [46].

Several research studies on the role of the Notch signaling pathway in liver cancer hint at its involvement in hepatocellular carcinoma (HCC) proliferation, invasion, organ metastasis, cancer cell stemness, and therapy resistance (Table 2).

Abnormal accumulation of Notch1, -2, -3, and -4 proteins has been detected in HCC tissues compared with surrounding normal liver or chronic hepatitis tissues [52,58,60]. Notch1 signaling exhibits controversial roles in hepatoma carcinogenesis, playing either oncogenic or tumor-suppressive functions. Ning et al. framed Notch1 activity as tumor-promoting, since its downregulation by curcumin, a known Notch1 inhibitor, led to cell growth arrest in vitro and significant suppression of HCC progression in vivo [47]. In contrast, Qi et al. demonstrated that Notch1 overexpression can inhibit both in vitro and in vivo HCC cell growth, through cell cycle arrest and apoptosis, probably induced by Notch1-mediated p53 upregulation and consequent Bcl-2 reduction [48]. In a later study conducted on primary liver cancer mouse models, it was reported that Notch1 inhibition reduced HCC-like tumors but strongly increased CCA-like nodules, underlining a possible highly context-dependent activity of Notch1 [39]. Aberrant expression of the Notch1 receptor has also been linked with EMT and HCC invasiveness. Wang et al. observed a significant association between high expression of Notch1 and metastatic disease, combined with increased Snail1 expression and decreased E-cadherin levels. In line with this, Notch1 knockdown in a mouse model reduced HCC metastasis [49]. Consistent with the previous finding, Giovannini et al. corroborated Notch1 activity in the EMT process through transcriptional and post-transcriptional regulation of the well-established EMT hallmark, E-cadherin. However, while Wang et al. reported E-cadherin suppression to favor tumor metastasis, in this study, Notch1 and E-cadherin expression levels were shown to positively correlate in human HCC tissues and promote invasiveness, revealing the complexity of the Notch1/Snail1/E-cadherin axis, as already highlighted by other works [49,50,61]. Furthermore, Notch1 accumulation was associated with an increased risk of HCC recurrence [50].

Several studies indicate that Notch2 signaling fosters tumor progression. Notably, Dill et al. demonstrated that constitutive Notch2 activation in HCC mouse models induces upregulation of pro-proliferative genes (cyclinD1 and cyclinA2), thereby increasing hepatocytes and biliary epithelial cell proliferation. In addition, they observed that constitutive Notch2 signaling expedites DEN-induced HCC development [51]. Consistent findings on the receptor were presented by Huntzicker et al., who showed that Notch2 inhibition can prevent tumor formation and decrease tumor burden in established primary liver cancers [39].

Beyond its essential role in cell proliferation, the involvement of Notch2 in carcinogenesis, progression, and drug resistance might be ascribable to its function in maintaining cell stemness [52]. Mounting evidence suggests that the presence of cancer stem-like features in the tumor is a key factor for HCC recurrence and therapy resistance [62].

A study by Wu et al. found that Notch2 knockout in hepatocellular carcinoma cell lines, HepG2 and SMMC-7721, causes a significant reduction in growth and self-renewal, and high expression of the receptor was observed in HCC cells positive for the hepatic stem cell marker CD90 [52]. Similarly, Hayashi et al. reported that Notch2-positive HCCs displayed more immature cellular morphology. Compared to Notch2-negative tumors, they were on average in more advanced clinical stages, with a significantly higher rate of positivity in metastatic HCCs [53].

The Notch3 receptor also appears to play an active role in HCC development. Investigation of Notch3 clinicopathological and prognostic significance in HCC revealed that high expression levels of the receptor were associated with more aggressive tumor traits: larger nodule size, multiple and later-stage tumors, and shorter overall survival [54]. A contribution of Notch3 signaling to drug resistance was highlighted by results obtained in Notch3-depleted HCC cells treated with doxorubicin. Notch signaling silencing combined with drug treatment enhanced doxorubicin uptake and HCC cell apoptosis via a p53-dependent mechanism [55]. The evidence that functional TP53 could be considered as a putative biomarker for selecting cases that might benefit from Notch3 inhibition was also confirmed by combining Notch3 inhibition with the tyrosine kinase inhibitor brivanib in different human cancer cells [56]. A similar study with sorafenib demonstrated improved efficacy of this multi-kinase inhibitor in HCC in vitro and in vivo following Notch3 inhibition, which promoted GSK3β phosphorylation and p21 downregulation [57].

Only a few studies have examined the role of Notch4 signaling in HCC. According to a comparative analysis of clinicopathological correlations of Notch receptor expression in HCC patients, abnormal accumulation of Notch4 can be an independent predictor of shorter disease-specific survival after curative resection, raising speculation of Notch4 involvement in tumor aggressiveness [58]. In line with these observations, another study by Cheng et al. identified a positive correlation between Notch4 expression and vasculogenic mimicry (VM) in HCC tissues, a phenomenon associated with high tumor grade, invasion, and metastasis in liver cancer [59,63]. They showed that downregulation of Notch4 hampered VM network formation and hindered cell migration and invasion in vitro and tumor growth in vivo [59]. This finding supports the important role of Notch4 in HCC tumor invasion.

## 5. Pharmacological Blockade of Notch Signaling in Cancer

Because the Notch signaling pathway plays a key role in inducing tumor growth, invasion, therapy resistance, and disease recurrence, it has increasingly been recognized as a promising druggable target for cancer treatment. Moreover, a number of studies have shown that downregulation of the Notch pathway increases the sensitivity of cancer cells to standard-of-care chemotherapeutic agents, thus making the pharmacological blockade of Notch signaling a useful approach to improve treatment outcomes in cancer patients [55,56,57,64,65].

In recent decades, multiple drugs have been designed to target various stages of the Notch signaling cascade, progressing from preclinical studies to clinical trials for both hematologic and solid malignancies.

γ-secretase inhibitors (GSIs) are a class of small molecules that target the γ-secretase complex, mediating the S3 proteolytic cleavage and inhibiting the release of the active Notch intracellular domain (NICD). Numerous preclinical studies have been conducted with GSIs in various tumor types, including NSCLC, colorectal cancer, hepatocellular carcinoma, prostate cancer, and breast cancer, showing impaired cell growth and tumor progression [12]. Given the encouraging results obtained in preclinical models, several GSIs, including BMS-986115 [66], LY900009 [67], MK-0752, LY3039478 [68,69,70], PF-03084014 [71,72,73], RO4929097 [74], and AL101 [75], have reached early-stage clinical trials [10,12]. However, only MK-0752 and PF-03084014 entered phase III/IV clinical trials. Most agents exhibited severe adverse events, with the major and dose-limiting side effect being gastrointestinal toxicity, likely an on-target toxicity due to the high constitutive expression of Notch in the GI tract and the important functions the pathway fulfills in the organ [76,77]. The observed adverse events caused by GSIs can be attributed to their pan-inhibitor nature. They non-selectively block the overall Notch signaling pathway by inhibiting γ-secretases, which also have other substrates besides the Notch receptors [76].

Nevertheless, currently, the only Notch-targeted therapy approved by the FDA for clinical use is nirogacestat (PF-03084014), a γ-secretase inhibitor that demonstrated significant progression-free survival (PFS) benefit over placebo in desmoid tumors [78].

To ameliorate the side effects associated with GSIs, γ-secretase modulators (GSMs) were also designed to modify the catalytic activity of specific γ-secretases without completely inhibiting their proteolytic function. MRK-560 is a selective inhibitor of PSEN1, an important catalytic subunit of the enzymatic complex, which was tested in preclinical studies on T-cell acute lymphoblastic leukemia (T-ALL) with activating Notch1 mutations. The drug effectively decreased the processing of Notch1 mutant and induced cell cycle arrest in vitro and in vivo, avoiding GI toxicity in animal models [79]. However, to date, only a few preclinical studies evaluating GSMs as anticancer agents are available, and no drugs have undergone clinical trials for human malignancies.

Another class of cleavage inhibitors was designed to block the metalloproteinases ADAM10 and ADAM17, responsible for the S2 cleavage that initiates downstream Notch signaling. A small-molecule ADAM inhibitor, INCB7839, was tested in early-phase clinical trials for solid tumors and breast cancer. The trials with this agent as monotherapy reported mild adverse events, but also restrictive toxicity, including deep vein thrombosis [80]. Another drug recently developed and classified as an ADAM inhibitor is a fully human monoclonal antibody against ADAM10 (1H5), which was shown to arrest the proliferation of colon cancer cell lines and CRC in mouse models. In addition, the therapeutical combination of 1H5 with irinotecan displayed significant tumor growth inhibition without visible signs of toxicity [81].

Alternative promising Notch inhibitor candidates are antibodies targeting Notch receptors and ligands. The use of antibodies as a therapeutic approach provides the benefits of a more target-specific treatment, reduced off-target effects compared to small-molecule drugs, and longer-duration response [82].

Diverse humanized Notch-specific antibodies against the receptors Notch1, Notch2, and Notch3 and the ligands Dll3 and Dll4 have been developed over the years. These macromolecules either block the receptor–ligand binding by targeting the EGF repeats or make the LNR region of the receptor inaccessible to the metalloproteases for the S2 cleavage [82]. The monoclonal antibodies Brontictuzumab (anti-Notch1), Tarextumab (anti-Notch2/3), PF-06650808 (ADC anti-Notch3), Rovalpituzumab tesirine (ADC anti-Dll3), SC-002 (ADC anti-Dll3), MEDI0639 (anti-Dll4), and Demcizumab (anti-Dll4) were all assessed in phase I/II clinical trials for solid tumors, either as monotherapies or, in some cases, as combinatorial treatments [10,80]. The majority of the studies resulted in a partial response, with drug-induced moderate antitumor activity or no improved in OS, PFS, or ORR. The side effects in most studies were manageable, and the antibody-based therapies were well tolerated by patients, except for a few cases of severe adverse events.

Monoclonal antibodies targeting the Notch pathway have yielded promising results in preclinical investigation as antitumor treatments; however, their translation into the clinic has faced some obstacles. The difficulty of the macromolecules in penetrating solid tumors, due to multiple kinetic barriers such as tumor vascular permeability and antibody internalization and clearance, often results in uneven distribution of the macromolecule within tumor tissue, which might affect therapeutic efficacy, leaving a fraction of cells untargeted [83,84]. Additionally, the antibody–drug conjugates (ADCs) can exhibit inadequate payload release, leading to off-target toxicity and low therapy potency [85].

Another noteworthy group of Notch-targeted drugs is the Notch transcription complex inhibitors—small molecules that target the NICD-CSL-MAML ternary complex, hindering its transcriptional function. A Notch transcription-blocking drug is CB-103, which showed positive preclinical results in endocrine-resistant BC and TNBC by inhibiting the formation of breast spheroids when combined with fulvestrant or paclitaxel [86]. The molecule has progressed to phase I/II clinical trials (NCT03422679) for advanced or metastatic solid tumors and hematological malignancies, which resulted in a manageable safety profile, although limited clinical antitumor activity as a monotherapy was observed [87] (Table 3).

With regard to liver malignancy, the research on the pharmacological blockade of Notch signaling has been limited to γ-secretase inhibitors and ADAM inhibitors in the preclinical setting [65,99,100]. To date, NCT03422679 is the first clinical study to include the investigation of a drug targeting the Notch signaling pathway in HCC [101].

Finally, a first-in-human study (NCT02722954) tested the combination of a Notch inhibitor, Demcizumab, with an ICI drug, pembrolizumab, in advanced or metastatic solid tumors. Although the results of this trial did not reveal significant antitumor activity from the treatment, further preclinical and early clinical studies are needed to overcome the challenges of mAbs and ADC in the clinics. Additionally, deeper investigation into Notch-inhibitor-based combinatorial therapies is warranted, given the multitude of studies highlighting the potentialities of Notch inhibitors as combinatorial drugs to reverse chemoresistance and inhibit tumor progression.

Since the pharmacological blockade of Notch signaling may be effective in eradicating mainly the sub-population of cancer cells with stemness properties, targeting the Notch pathway alone appears unsatisfactory for the eradication of bulk cancer cells [28]. The combination of Notch-targeted drugs with immune checkpoint inhibitors, antiangiogenic agents, or chemotherapy holds the promise of enhancing synergistic therapeutic effects. Continued research in this area is essential for unlocking the full potential of Notch-targeted therapies in cancer treatment.

## 6. Notch Signaling in the Tumor Microenvironment

Cancer characteristics and behavior rely on the bidirectional interaction between cancer cells and the tumor microenvironment, forming a dynamic and complex network. The crosstalk between cancer cells and their environment involves various signaling pathways, including the Notch pathway. Indeed, through paracrine and juxtracrine signals, Notch regulates many features of the TME, including immune infiltrate functions [102].

Notch signaling recruits and induces macrophage polarization into the M2 type through the secretion of IL-4 and IL-6, leading to breast cancer development [103]. In line with this, Notch-deficient macrophages are less effective in reducing the proliferation of B16 or LLC1 syngeneic grafts compared to wild-type macrophages [104]. Consistent with the previous finding, patients with mutations in the Notch pathway exhibited an inflammatory TME characterized by greater infiltration of activated immune cells, including M1 macrophages, CD8+ T cells, neutrophils, and NK cells, as well as enhanced immunogenicity [105,106]. Following recruitment, Notch-activated macrophages suppress the proliferation of tumor-infiltrating T cells and their tumor-killing activity by enhancing CD14 and CD93 secretion [107]. Notch1 inhibits the intratumoral infiltration of natural killer (NK) cells and CD8+ cytotoxic T lymphocytes while increasing Tregs and MDSCs through TGF-β1 upregulation, leading to melanoma proliferation [108]. Notch1 signaling also facilitates chronic lymphocytic leukemia (CLL) escape from immune surveillance by blocking antigen presentation and inhibiting T-cell activation [109]. High Notch3 expression correlates with decreased infiltration of activated CD8+ T cells and the emergence of immunosuppressive cells (e.g., Tregs and M2 macrophages).

Other evidence suggests a tumor-suppressive role of Notch signaling within TME [109]. For instance, in gastric cancer, the upregulation of Notch receptors has been associated with greater infiltration of CD4+ T cells, DCs, neutrophils, and macrophages [110]. In glioma, Notch signaling has been reported to favor the recruitment of antitumor cytotoxic T lymphocytes and to promote the conversion of TAMs into the M1 antitumorigenic phenotype. Notably, Parmigiani et al. have demonstrated that the suppression of the Notch pathway enables the tumor to evade immune surveillance [111] (Table 4).

The dual role of the Notch signaling pathway in cancer is associated with its capacity to regulate both tumor cells and immune cells in the tumor microenvironment, influencing their complex interactions. However, the conditions and the factors that affect the outcome of Notch signaling—either tumor-promoting or tumor-suppressive—remain poorly understood.

Based on these studies, Notch inhibition has a strong impact on the activation of immune cells in the microenvironment. Further research is necessary to determine whether these effects are an indirect consequence of tumor growth inhibition. Hence, experiments using in vitro co-culture systems with immune and cancer cells may be of interest in this regard.

The tumor microenvironment plays a pivotal role in the etiopathogenesis of HCC and in tumor aggressiveness, including EMT, invasion, and metastatization [112].

Activated TAMs increase angiogenesis and hepatocarcinogenesis. Moreover, TAM density is associated with advanced tumor stage, disease recurrence, and poor overall survival in HCC patients [113].

Specifically, the HCC microenvironment displays features reminiscent of fetal development, including fetal-like tumor-associated macrophages. Human fetal liver and HCC share a transcriptional program involved in maintaining an immunosuppressive oncofetal ecosystem, as well as spatial transcriptomics. In vitro functional assays have demonstrated the involvement of VEGF and Notch signaling in maintaining this oncofetal ecosystem [114]. This concept of a VEGF/Notch-mediated immunosuppressive oncofetal ecosystem opens the possibility of a new combination target with immunotherapy.

## 7. The PD-1/PD-L1 Pathway

Programmed cell death protein 1 (PD-1, also referred to as CD279) is a type I transmembrane protein and a member of the CD28/CTLA-4 extended family, encoded by the PDCD1 gene. It consists of 288 amino acids arranged into an extracellular IgV-like domain flanked by a signal sequence and a stalk region, a transmembrane domain, and an intracellular cytoplasmatic tail that harbors two tyrosine-based signaling motifs: the immunoreceptor tyrosine-based inhibitory motif (ITIM) and the immunoreceptor tyrosine-based switch motif (ITSM). Both ITIM and ITSM are essential for transducing the PD-1 inhibitory signal [115].

Programmed cell death-ligand 1 (PD-L1, also known as CD274) and ligand 2 (PD-L2, CD273) are the main ligands of PD-1. They are both type I transmembrane proteins and products of the CD274 and PDCD1LG2 genes, respectively [115]. These ligands exhibit a similar structural organization, with an Ig-V and Ig-C-like extracellular domain, a membrane-permeating domain, and a short cytoplasmatic tail that features three non-classical signaling motifs: the RMLDVEKC, DTSSK, and QFEET sequences, which are involved in PD-L1 stability regulation and signal transduction [116,117,118].

The PD-1/PD-L1 axis has evolved as an immune checkpoint pathway to control the magnitude and duration of immune responses, minimizing tissue damage and promoting self-tolerance to prevent autoimmune reactions [119]. Notably, ligand binding to PD-1 on immune cells induces inhibitory responses. In T lymphocytes, the PD-1/PD-L1 axis delivers an intracellular inhibitory signal that can dampen T-cell effector functions and decrease proliferation and survival [120].

The interaction of PD-1, expressed by activated T lymphocytes, with its ligands, PD-L1 and PD-L2, on APCs induces a conformational change in PD-1 and subsequently the phosphorylation of the two tyrosine-based signaling motifs in the cytoplasmatic region, carried out by Src family kinases [118]. This event triggers the recruitment of SHP-2 tyrosine phosphatase, which can attenuate T-cell activation by dephosphorylating key signaling elements. Studies have found that the primary target of PD-1-engaged SHP-2 phosphatase is the costimulatory receptor CD28, rather than the TCR, suggesting that PD-1 halts T-cell activation primarily by hindering CD28 signaling [121].

The PD-1 receptor is transiently expressed on all T cells upon antigen-mediated activation through the TCR. Following antigen clearance, PD-1 levels on responding T cells decrease. However, persistent antigen exposure and/or inflammation, such as in chronic infections and cancer, elicits a constitutive high and sustained PD-1 expression, which can progressively lead to “exhausted” T lymphocytes with impaired effector functions [120,122,123] (Figure 2).

Several other immune cell populations express PD-1, including activated B cells, natural killer (NK) cells, macrophages, monocytes, dendritic cells (DCs), and myeloid precursor cells. In addition, PD-1 can be detected in cancer cells [124].

The expression of PD-1 ligands extends widely to non-lymphoid tissues [119]. PD-L1 is expressed in diverse cell types, both from the hematopoietic lineage, including T cells, B cells, DCs, macrophages, and mast cells, and from the non-hematopoietic lineage, such as some epithelial and endothelial cells. Conversely, PD-L2 distribution is more restricted and only detectable in APCs and mast cells. Both ligands are expressed by various tumor cells and tumor stroma, with PD-L1 being the predominant one [118,120].

In the context of cancer, the PD-1/PD-L1 axis is exploited to dampen the antitumor immune response. Neoplastic cells frequently hijack host surveillance by overexpressing PD-L1, taking advantage of peripheral immune tolerance, leading, in case of chronic stimulation, to the physiologic limitation of the immune response to prevent autoimmune phenomena by developing constitutive PD-1^high^ dysfunctional T lymphocytes, which are unable to restrain tumor progression [122].

The aberrant overexpression of PD-L1 in the tumor microenvironment (TME) can be caused by an alteration of the complex regulatory network that controls PD-L1 levels. CD274 undergoes epigenetic and transcriptional regulation. Promoter DNA methylation and histone modifications have been shown to modulate PD-L1 expression [117]. Interleukins and several inflammatory cytokines, including IFN-α, IFN-β, IFN-γ, and TNF-α, induce CD274 transcription, with IFN-γ considered one of the most potent soluble drivers of PD-L1 expression [118,125]. The activation of cell-intrinsic oncogenic signaling can also increase PD-L1 expression. Multiple oncogenic transcription factors, such as MYC, STAT3, IRF1, and NF-kB, have been reported to directly modulate CD274 transcription [125]. Hypoxia, as a stressor factor, affects PD-L1 expression by prompting HIF-1a and HIF-1b production, which subsequently upregulates gene transcription [118]. In addition, PD-L1 levels are influenced by microRNA-based control of PD-L1 messenger and post-translational modulation [125].

Likewise, the expression of the receptor PD-1 is subject to intricate regulation at both the genetic and epigenetic levels, through DNA methylation and histone modification, in response to transient or persistent antigen stimulation [126]. Transcription factors such as NFATc, FOXO1, IRF9, and AP-1 can induce PD-1 expression by binding the PDCD1 gene promoter [127]. Furthermore, Mathieu et al. demonstrated that Notch signaling regulates PD-1 expression by direct binding of the NICD-CSL complex to the promoter [128]. Conversely, T-bet and Blimp-1 have been identified as inhibitory factors of CD279 transcription [116].

## 8. PD-1/PD-L1 Signaling in Epithelial Cells

The PD-1/PD-L1 pathway is a crucial regulator of immune activation and plays a key role in mediating peripheral tolerance. While the cytotoxic T-lymphocyte-associated antigen 4 (CTLA-4) is known to act as an immune checkpoint in the initial stage of naïve T-cell activation, primarily in lymph nodes, the PD-1 pathway intervenes at later stages of the immune response, in previously activated T lymphocytes, in the peripheral tissues where PD-1 ligands are typically expressed [129]. It is well established that the PD-1/PD-L1 axis is used by cancer cells as an immune escape mechanism. By increasing the expression of immune checkpoints and reducing the levels of immune costimulatory molecules, neoplastic cells shield tumors from T-cell-mediated killing, mainly inhibiting the cytotoxic T-lymphocyte effector functions [117,130]. In several tumor types, such as liver, breast, ovarian, gastric, colorectal, lung, and pancreatic cancer, elevated expression of PD-L1 has been reported, and it is generally correlated with an unfavorable prognosis [131,132]. However, a number of studies have shown that PD-L1 expression could also hold positive prognostic value in some carcinomas, thus highlighting the controversial role of PD-1 ligands as predictive and prognostic biomarkers for clinical outcomes. For instance, in NSCLC and colorectal cancer, conflicting findings have emerged, identifying both positive and negative prognostic roles for tumor-associated PD-L1 [131]. Similarly, in the liver, Gao et al. reported that HCC patients with higher expression of PD-L1 had significantly poorer disease-free survival (DFS) and overall survival (OS) after curative HCC surgery [133]. This study was confirmed by Jung et al., who demonstrated a strong correlation between upregulated PD-L1 and simultaneous overexpression of both PD-1 ligands, as well as worse prognosis [134]. In contrast, research by Xie et al. pointed to PD-L1 upregulation as a favorable prognostic factor for HCC, as ligand expression was strongly associated with superior DFS and OS outcomes [135].

This controversial evidence could result from the complex regulatory mechanisms of PD-L1 in cancer cells, which are dependent on the context-specific underlying transcriptional and signaling network.

In some tumors, PD-L1 upregulation is a mechanism of adaptive response to antitumor immunity. The expression of the ligand is induced by inflammatory cytokines, especially INF-gamma, released by intratumoral immune cells. In this setting, PD-L1 is not constitutively and uniformly expressed, but it predominantly characterizes tumor regions with high lymphocyte infiltration [136].

In other cases, aberrant activation of oncogenic signaling could lead to constitutive PD-1 ligand expression on neoplastic cells, independently of inflammatory signals from the tumor microenvironment, a mechanism of innate immune resistance [136]. PD-1/PD-L1 signaling crosstalks with multiple oncogenic pathways, including PI3K/AKT/mTOR, MAPK, JAK/STAT, WNT, NF-kB, Notch, and the Hedgehog (Hh) pathway, which promote the expression of the immune checkpoint axis [127,128]. Activating alterations in these pathways can promote tumor immune evasion by upregulating PD-L1 expression. PI3K mutations or PTEN loss in NSCLC, squamous cell lung carcinoma, breast, prostate, and colorectal cancers results in constitutive AKT/mTOR activation, which consequently augments PD-L1 production. Similarly, mutations of RAS, BRAF, and EGFR that elicit PD-L1 expression via MAPK pathway activation have been identified in some tumors [117].

Moreover, PD-L1 genetic copy number gain or chromosomal amplification can be responsible for PD-1 ligand overexpression, an event observed in cancers such as TNBC and squamous cell carcinoma of the vulva and cervix [116].

Antibody-mediated blockade of the PD-1/PD-L1 axis is a successful ICI immunotherapy used in many advanced solid tumors as a first- or second-line treatment, either in monotherapy or in combination [137]. These immunomodulating agents stimulate the immune system to reactivate the antitumor response, primarily by re-invigorating exhausted T cells and prompting tumor T-cell killing [115]. ICI therapy has demonstrated great efficacy, especially in non-small-cell lung cancer (NSCLC) [138], melanoma [139], and hepatocellular carcinoma (HCC) [140], with a longer duration of response and an improved toxicity profile compared to traditional or alternative treatments. Nevertheless, a large percentage of patients show no clinical response to the immune treatment or even hyperprogressive disease, events indicative of primary (intrinsic) resistance and secondary (acquired) resistance [115].

Besides the conventional extrinsic role of the PD-1/PD-L1 axis in inhibiting the immune system—exploited by neoplastic cells, contributing to acquired resistance—recent studies have revealed a tumor-intrinsic role of PD-1 and PD-L1 that might further affect therapy response [117]. PD-1 expression has been detected in different cancer cell types, including ovarian, breast, pancreatic, renal, lung, hepatic carcinoma, and colorectal cancer cells, with differential tumor-specific roles [117,141]. In liver cancer cells, intrinsic-PD-1 has been reported to promote tumorigenesis through the mTOR pathway. Blockade and knockdown of PD-1 inhibited tumor growth in vitro and in mouse xenograft models [142]. In TNBC, Wu et al. demonstrated that PD-1 in neoplastic cells enhanced tumor growth and metastasis, both in vitro and in vivo [143]. Conversely, in NSCLC, intrinsic PD-1 exhibited an antitumor role by downregulating signaling pathways such as PI3K/AKT and MAPK/ERK1/2, and its silencing or antibody blockade resulted in accelerated proliferation via PI3K and MAPK pathway activation in both NSCLC and colon cancer cells [144,145].

Similarly, the emerging intrinsic role of PD-L1 signaling in tumor cells has been found to be protumorigenic in various types of neoplasia [146,147,148,149] while tumor-suppressing in cholangiocarcinoma [117]. The mechanism of PD-L1 upregulation in tumor cells in response to interferons not only hampers T-cell effector functions but also forms an anti-apoptotic shield. This shield increases resistance to IFN-mediated apoptotic stimuli through reverse signaling that inhibits STAT3. Furthermore, intrinsic PD-L1 activates the cancer cell inflammasome pathway, which triggers the recruitment of myeloid-derived suppressor cells (MDSCs) and boosts the establishment of an immunosuppressive tumor microenvironment (TME) [115].

Accumulating evidence has highlighted the involvement of PD-L1 in several other pro-carcinogenic mechanisms, including cancer stemness, tumor invasion, metastasis, and therapy resistance [117]. For instance, Li et al. reported that, in human gastric cancer cell lines, PD-L1 knockdown inhibited cell proliferation, migration, and invasion while increasing sensitivity to CIK therapy and reducing tumor growth and the EMT phenotype in mouse models [149]. Interestingly, the immune checkpoint genes CD274, CTLA4, CD276, and CD200 are upregulated by Notch3 in gastric cancer, leading to impaired immune responses [150].

In MDA-MB-231 breast cancer cells, Liu et al. demonstrated that PD-L1 upregulates the expression of multidrug resistance 1/P-glycoprotein (MDR1/P-gp) through the activation of PI3K/AKT and MAPK/ERK pathways [148], thereby inducing chemotherapy resistance. Similarly, Chen et al. showed that PD-L1 promotes EMT in MDA-MB-231 cells via the p38-MAPK/Snail pathway and PD-L1 deficiency attenuates the lung metastasis of TNBC in vivo [151] (Table 5).

Finally, more recent findings indicate that PD-L1 can regulate the expression of multiple immune response genes by directly binding to DNA, following an acetylation-dependent translocation from the plasma membrane to the nucleus [152].

To date, there are limited studies examining the effect of immunotherapeutic antibodies on intrinsic pathways of PD-1 and PD-L1 in cancer cells. Existing research suggests that ICIs may be able to modulate the intrinsic roles of PD-L1 and its receptor. However, further investigation is necessary to fully understand how immunotherapy impacts the PD-1/PD-L1 intracellular tumoral axis and whether the co-expression of PD-1 and PD-L1 in cancer cells contributes to drug resistance [117].

## 9. PD-1/PD-L1 Signaling in Immune Cells

Based on the physiological function of the PD-1/PD-L1 pathway in tuning the immune response to maintain immune homeostasis and prevent autoimmunity, PD-1 and its ligands mediate immune suppression in both the adaptive and innate immune systems by regulating the activation and function of different immune cell types. PD-1 expression on naïve T cells is induced upon antigen encounter, TCR stimulation, and CD28 co-stimulation signaling. Initially, PD-1 expression reflects T-cell activation, with PDCD1 transcription levels indicating the strength of TCR signaling and the avidity of specific T cells. However, when ligands expressed by tumor cells or immune infiltrating cells in the TME bind to PD-1, an intracellular inhibitory signal is transduced that halts the activation process and impairs T-cell effector functions [122]. In the event of chronic antigenic exposure, persistent PD-1 signaling can prompt a distinct pattern of cellular differentiation, characterized by the increased co-expression of inhibitory receptors, such as TIM-3, LAG-3 or TIGIT, reduced cytokine production, and altered glycolysis and mitochondrial metabolism. These changes lead to T-cell dysfunction and exhaustion [153].

Similarly, T lymphocytes upregulate PD-L1 expression in response to antigen stimulation and inflammatory signals. Diskin et al. unveiled PD-L1 back-signaling in T cells: upon ligation, PD-L1 induces intracellular suppressing signaling similar to that of PD-1. Additionally, PD-L1+ T cells have been found to inhibit neighboring effector T cells through canonical PD-1/PD-L1 interactions and promote STAT6-dependent M2-macrophage differentiation [154].

The PD-1/PD-L1 axis is also involved in the developmental fate of CD8+ central memory T cells. PD-1 signaling negatively regulates the differentiation of CD8+ central memory T cells into CD8+ effector memory T cells, thus influencing immunological memory [155].

In B cells, the PD-1/PD-L1 pathway similarly inhibits the BCR-mediated activation by recruiting SHP-2 tyrosine phosphatase, which dephosphorylates downstream BCR signaling molecules, resulting in altered CA^2+^ cellular concentration and long-term growth arrest. Moreover, PD-1 activation impairs antibody secretion in response to type 2 antigen stimulation [156].

Natural killer (NK) cells, important effector lymphocytes of the innate immune system that participate in antitumor responses by directly killing tumor cells or recruiting T cells, are also negatively regulated by PD-1. This receptor serves as a checkpoint for NK activation. It is highly expressed on activated NK cells with a more responsive phenotype but, upon interaction with PD-L1 or PD-L2, inhibits NK cells’ degranulation, thus hindering their cytotoxic functions [153,157].

Another population involved in the innate immune response is neutrophils, which exhibit both pro- and antitumor roles. Neutrophils in the TME express high levels of PD-L1 and display tumor-promoting properties, as they have been shown to suppress T-cell functions [158]. PD-L1+ infiltrating neutrophils are considered poor prognostic indicators in certain human tumors. For instance, Wang et al. observed that activated PD-L1+ neutrophils contributed to T-cell suppression in vitro and to gastric cancer growth and progression in vivo [159].

In addition, PD-L1 has been shown to inhibit the antitumor cytotoxicity of neutrophils [158].

In dendritic cells (DCs), PD-L1 plays a critical role in regulating T-cell responses. The expression of this ligand is upregulated following antigen uptake, preventing the excessive expansion of activated tumor-infiltrating lymphocytes by engaging PD-1 on T cells. Furthermore, it protects APCs from being targeted and killed by activated cytotoxic T lymphocytes [160].

The PD-1/PD-L1 axis fosters an immunosuppressive environment also by promoting the generation of regulatory T cells (Tregs) and the polarization of tumor-associated macrophages (TAMs).

Specifically, in regulatory T cells, the key mediators of peripheral tolerance, PD-1 and PD-L1 expression induces the transformation of naïve CD4+ T cells into Tregs and enhances their activity within the TME [153].

TAMs, the most abundant immune cell population in the TME, highly express PD-1 and PD-L1. These immune checkpoints have been shown to deliver a negative signal that modulates the TAM phenotype and functions, shifting them towards an immune-suppressive, protumorigenic M2 macrophage type. In addition, recent studies revealed that PD-1 can weaken the phagocytic ability of TAMs against tumor cells and their cytokine production [161].

As a tumor progresses, the accumulation of M2-polarized TAMs has been observed, alongside an increased expression of both PD-1 and PD-L1, populating the immunosuppressive niche that facilitates tumor escape [161] (Table 6).

PD-1/PD-L1 blocking immunotherapy not only impacts cytotoxic T cells and cancer cells but also affects multiple targets among tumor-infiltrating immune cells, which collectively define the dynamic balance of the tumor immune system. Therefore, it is of uttermost importance to consider the role of this axis across various immune cell types to achieve a deeper and more comprehensive understanding of patient responses to immunomodulatory treatments.

## 10. The Crosstalk Between the Notch Pathway and PD-1/PD-L1 Axis

Notch signaling contributes to the carcinogenesis of different types of tumors through multiple mechanisms, such as cancer stem cell self-renewal, EMT, and drug resistance. Additionally, recent studies have highlighted the pivotal role of Notch signaling in modulating the immune tumor microenvironment. This pathway can influence the activity of various immune cells, including T and B cells, macrophages, DCs, NKs, neutrophils, MDSCs, and CAFs [109].

Similarly, the PD-1/PD-L1 axis within the tumor microenvironment plays a crucial role in supporting cancer progression by neutralizing the immune system and, as a tumor-cell-intrinsic signal, by regulating cell proliferation and stemness, while also promoting invasion, metastasis, and resistance to anticancer therapies.

Growing evidence has revealed the crosstalk between Notch and PD-1/PD-L1 signaling pathways.

Both pathways are part of a broader communication network involving other stem-cell-associated signaling routes, and they feature interconnection points with each other. Notch signaling directly regulates the transcription of the PDCD1 gene in activated CD8+ T cells via the binding of the NICD-CSL complex to the promoter. In addition to Notch’s role in T-cell activation, there is speculation that it contributes to cytotoxic lymphocyte exhaustion by maintaining high PD-1 expression levels [128]. Inhibition of Notch signaling in tumor-infiltrating CD8+ T cells from colorectal carcinoma patients enhanced the cytotoxic activity of these lymphocytes and was accompanied by reduced PD-1 receptor expression [162]. These findings by Yu et al. support the hypothesis that the Notch pathway might be involved in CD8+ T-cell dysfunction.

Furthermore, the Notch pathway is deemed responsible for PD-L1 upregulation in gastric cancer cells, thereby favoring immune escape. According to Jiang et al., in ubiquilin-4-high gastric cancers, the overexpressed protein promotes the degradation of Numb—a negative regulator of NICD—thus activating the Notch pathway and enhancing PD-L1 expression [163].

Consistent findings demonstrated that Notch signaling indirectly regulates the PD-L1 expression in breast cancer stem-like cells by inducing ligand overexpression through mTOR activity [164] (Figure 3).

Likely due to the crosstalk with the PD-1/PD-L1 axis, the Notch pathway has been described as a factor influencing the prognosis of antitumor immunotherapy in many types of cancer. For instance, mutations in Notch signaling have been associated with enhanced efficacy of immune checkpoint inhibitors in NSCLC and CRC. NSCLC and CRC patients with high-mutated Notch signaling after receiving ICIs showed a favorable prognosis with increased PFS and OS, which was found to be related to enhanced tumor immunogenicity and a greater number of activated TILs, proinflammatory factors, and higher expression levels of immune checkpoint molecules within TME [105,106].

Altogether, these findings contribute to a better understanding of the central role of Notch signaling in therapy resistance and suggest that the Notch pathway is a promising druggable target for anticancer treatment, potentially complementing conventional therapeutic options to prevent intrinsic or acquired resistance.

Recent studies have investigated the use of ICI-based therapy in combination with selective Notch inhibitors in a preclinical setting [165]. Specifically, Dai et al. have tested the co-administration of a Notch inhibitor (DAPT) and an anti-PD-L1 antibody in pancreatic cancer mouse models, which resulted in significant inhibition of tumor growth compared to monotherapy, induced PD-L1 overexpression, and increased number of tumor-infiltrating CD8+ T cells [165]. Likewise, Meng et al. have shown that in Jagged1-high TNBC, the inhibition of Notch signaling by a γ-secretase inhibitor combined with an anti-PD-1 antibody markedly delayed tumor growth in vivo [107].

Although the number of studies available is limited and further investigation is needed, these findings are promising and uphold the rationale for the combination of ICI with selective Notch inhibitors as an anticancer treatment [165].

To date, preclinical and clinical studies investigating the crosstalk between Notch signaling and PD-1/PD-L1 axis in the liver are lacking. However, evidence obtained from different cancers suggests a potential interaction between Notch and PD-1/PD-L1 pathways in the liver, where they might contribute to the development and progression of HCC.

Further research is required to shed light on their potential interplay, elucidate the mechanisms that could contribute to HCC onset and treatment resistance, and assess the effects of ICI-based combination therapies in hepatocellular carcinoma.

## 11. Discussion

Immunotherapy has been shown to improve both the lifespan and quality of life of patients with advanced HCC. However, current treatment options still have a relatively low success rate of about 30%. The clinical application of PD-1/PD-L1 blockade therapies was initially thought to prevent tumor progression by “removing the breaks” on T cells. However, other factors beyond “removing breaks” might influence the efficacy of PD-1/PD-L1 blockades, potentially explaining why some patients achieve complete responses while others show no response. The Notch signaling pathway is an evolutionarily conserved pathway that influences multiple lineage decisions of developing lymphoid and myeloid cells and serves as an important modulator of T-cell-mediated immune responses [166]. In line with this, a recent study demonstrated that Notch3 signaling promotes PD-L1 overexpression in breast cancer cells, and specific Notch3 silencing results in PD-L1 downregulation, providing a novel strategy for anti-PD-L1 combination therapies [164]. We previously showed that breast cancer and HCC share a common Notch3 molecular signature, leading us to hypothesize that Notch inhibition may also enhance the effects of atezolizumab in advanced HCC [56].

Although Notch receptor expression has been widely described in hepatocarcinoma cells [52,58,60], biomarkers related to its activation status need to be identified to stratify patients scheduled for immunotherapy. Villanueva et al. provided evidence that Notch signaling is involved in the pathogenesis of HCC, with 30–35% of tumors exhibiting pathway activation based on a well-characterized gene signature [167]. However, the lack of HCC tissue samples highlights the need to identify circulating biomarkers of Notch signaling activation. In this perspective, Giovannini et al. identified soluble E-Cadherin, Thbs1, and Pai3 as putative serum biomarkers of Notch1 activation in vitro and in human HCC [50]. There remains an unmet need for additional serum biomarkers to better select patients with Notch pathway activation and, even more so, to identify the exact source of these soluble molecules. Indeed, Notch signaling is known to play an essential role in tissue renewal and maintenance across various organs, including the nervous system, kidney, liver, skin, muscle, and bone [27]. This widespread role may explain why Notch silencing through Gamma secretase inhibitors (GSIs) induces toxicity. To address this, we observed a more specific delivery of Notch3 siRNA to the liver using double-stranded RNA-mediated interference (RNAi), potentially avoiding major complications [56]. HCC patients secrete higher levels of exosomes enriched with molecules that could facilitate understanding of Notch signaling activation, warranting further investigation. The detection of Notch receptors as well as downstream targets in exosomes enriched with specific HCC proteins, including α-fetoprotein (AFP) and asialoglycoprotein receptor 1 (ASGPR1), could guide the identification of patients eligible for combined treatments with immune checkpoint inhibitors.

Since exosomes can transport various biomolecules derived from their parent cells, tumor-derived exosomes are challenging in clinical settings. However, validation steps are required to use exosomes as a source of tumor biomarkers. First of all, it would be necessary to have a standardized method for isolating exosomes that can provide substantial yields and a reliable quality of exosomes. Second, tumor-derived exosomes would need to be enriched with the help of specific surface biomarkers such as ASGPR1 for HCC. ASGPR1-positive and -negative exosomes should be sorted and analyzed for the expression of other HCC-specific genes, including Albumin and α-fetoprotein, thereby aiding in establishing the robustness of the membrane marker in identifying exosomes released from the tumor. Notably, Notch expression could be analyzed in ASGPR1-positive exosomes, and cancer could be simply categorized as positive or negative for Notch expression. Remarkably, for many tumors, most clinical trials are conducted with patients for whom the possession status or expression level of the target is unknown [168].

## 12. Conclusions

The Notch signaling pathway has emerged as a crucial player in cancer development and progression, revealing pivotal roles in cancer stem cell stemness, angiogenesis, invasion, and organ metastasis. Significantly, the involvement of Notch signaling in drug resistance makes the pathway a promising druggable target to prevent tumor resistance and improve the clinical outcomes of cancer patients. Solely targeting the Notch pathway has been proven insufficient for the eradication of bulk cancer cells; however, the combination of Notch inhibitors with ICIs has yielded encouraging results in the preclinical settings, providing a promising strategy for the treatment of HCC.

Further research and innovation are essential to assess the potentialities of Notch-targeted therapy combined with ICIs in cancers. Additionally, proper selection of patients who may benefit from combination therapy is needed; the analysis of tumor-derived exosomes expressing Notch could guide the stratification of patients with liver malignancy who are eligible for Notch inhibition in combination with ICI-based therapy.

## Figures and Tables

**Figure 1 biomolecules-14-01581-f001:**
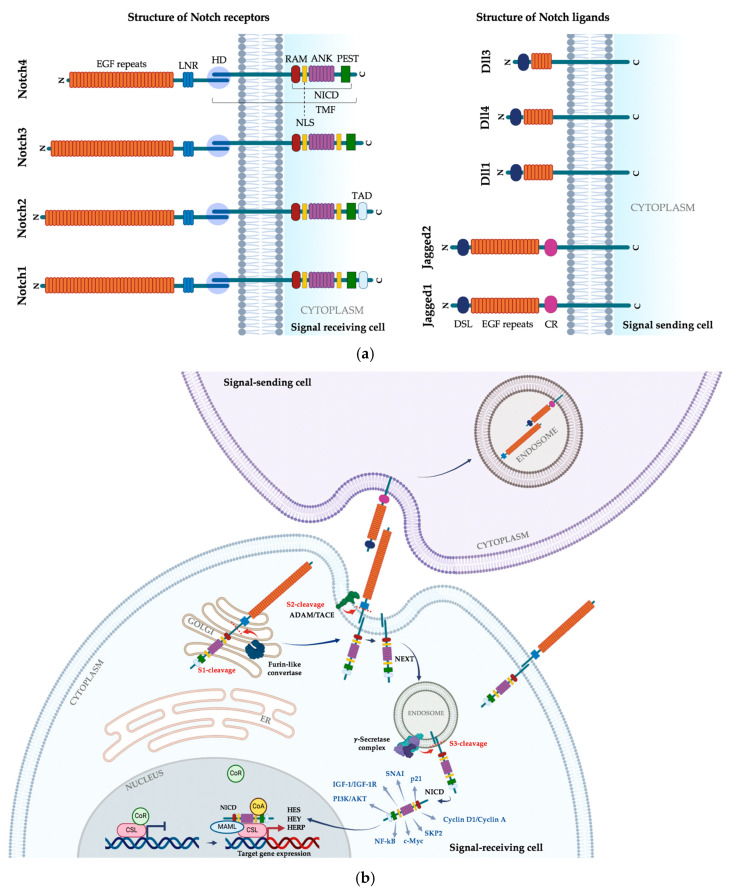
Schematic representation of canonical Notch signaling pathway. (**a**) Schematic representation of the structure of Notch receptors and ligands. (**b**) Each Notch receptor is synthesized as a single precursor protein, cleaved into a heterodimer within the Golgi apparatus, and transported on the cell surface. Upon ligand binding on the neighboring cell, the receptor undergoes consecutive cleavages by TACE and γ-secretase, which release the Notch intracellular domain (NICD). NICD translocates into the nucleus, where it interacts with the transcription factor CSL, displaces co-repressors (CoRs), and recruits co-activators (CoAs), such as MAML. This process results in the activation of downstream target genes belonging to the HERP, HES, and HEY families. Figure was created with BioRender.com.

**Figure 2 biomolecules-14-01581-f002:**
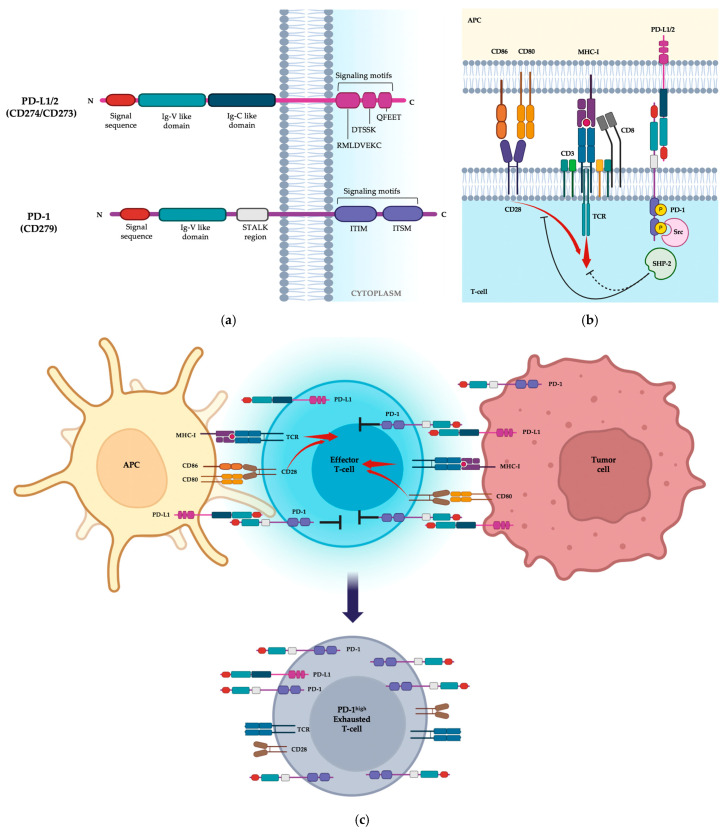
PD-1/PD-L1 signaling pathway. (**a**) PD-1 and PD-L1 structures consist of an extracellular domain, a transmembrane domain, and an intracytoplasmic region (signal motifs). (**b**) Antigens presented by Antigen-Presenting Cells (APCs) are recognized by the T-cell receptor (TCR; first signal). The second signal is delivered when CD80 and CD86 on the APCs engage CD28 on the T cells (red arrows indicate the activation signals, black and dotted arrows represent the suppression of the activation process). (**c**) The first signal for T-cell activation is provided by the binding of the T-cell receptor (TCR) with the antigen (Ag) presented by the APC in the context of the Major Histocompatibility Complex (MHC). A cascade of events follows, triggering the proliferation of effector T cells. Upon antigen-mediated activation (red arrows), the PD-1 receptor is transiently expressed on T cells and interacts with Programmed death-ligand 1 (PD-L1) overexpressed on the surface of APCs or cancer cells, suppressing the immune response (black arrows). Following antigen clearance, the PD-1 levels on responding T cells decrease. Persistent antigen exposure and/or inflammation, such as in chronic infections and cancer, elicits a constitutive high and sustained PD-1 expression, which can progressively lead to “exhausted” T lymphocytes with impaired effector functions. Figure created with BioRender.com.

**Figure 3 biomolecules-14-01581-f003:**
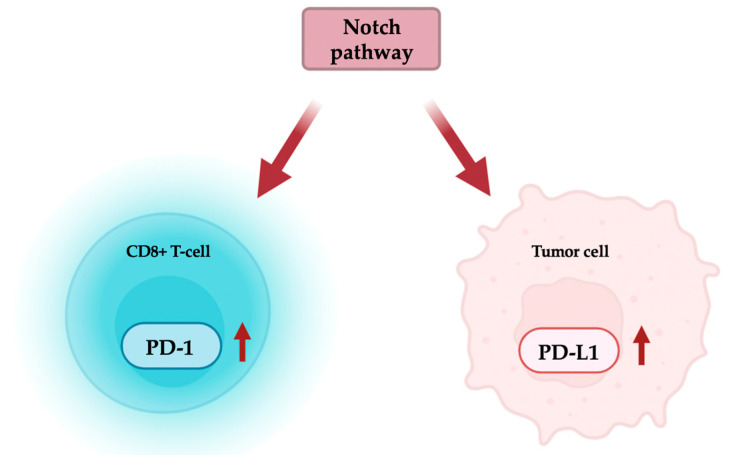
Schematic representation of the known functional interactions between Notch and PD-1/PD-L1 pathways. Notch signaling regulates the expression of PD-1 in CD8+ T lymphocytes, promoting PDCD1 transcription (depicted by the red arrow) by the direct interaction of the NICD-CSL complex with the PDCD1 promoter. In tumor cells, the Notch pathway determines the upregulation of PD-L1 ligand expression (red arrow). However, the exact mechanism of regulation needs to be elucidated further. Figure created with BioRender.com.

**Table 1 biomolecules-14-01581-t001:** Divergent roles of Notch signaling in cancer.

Notch Signaling Deregulations	Cancer Type	Putative or Observed Effect	References
Activity-enhancing mutations of Notch1 and Notch2	TNBC	Tumorigenesis	[30]
Activity-enhancing mutations of Notch1	NSCLC	Tumorigenesis	[31]
Activity-reducing mutations of Notch1, -2, -3, and -4	SCLC	Tumorigenesis	[32]
Activity-reducing mutations of Notch1, -2, -3	HNSCC, ESCC, cSCC, and lung squamous cell carcinoma	Tumorigenesis	[28]
Dll4/Notch4 upregulation	Lung adenocarcinoma	EMT/tumor progression	[33]
Notch3 upregulation	ERα+ and TNBC	EMT/tumor metastasis	[34,35]
Notch1/Jagged1 upregulation	Metastatic prostate cancer (PCa)	PCa progression and metastasis	[36]
Notch4-HEY1 upregulation	HNSCC	EMT induction	[37]
Notch3 downregulation	ESCC	EMT induction	[38]
Notch2/Jagged1 upregulation	Pancreatic cancer (PC)	EMT, chemoresistance phenotype	[40]
Notch1 knockdown or inhibition/ Notch4 inhibition	Breast cancer (BC)	BCSC reduction	[41]

**Table 2 biomolecules-14-01581-t002:** Roles of Notch signaling in hepatocellular carcinoma.

Notch Receptor Involved	Role in HCC/Involved Biological Process	Putative or Observed Effect	Reference
Notch1	Oncogene	Notch1 inhibition induces growth arrest in vitro and in vivo	[47]
	Oncogene	Notch1 inhibition reduces HCC-like tumors	[39]
	Tumor suppressor	Overexpression of Notch1 arrests cell growth in vitro and in vivo	[48]
	Oncogene/EMT	Knockdown of Notch1 reduces HCC metastasis	[49]
	Oncogene/EMT	Notch1 regulates E-cadherin, stimulating invasiveness	[50]
	Oncogene/associated with HCC recurrence	Notch1 accumulation in liver tissue is associated with increased risk of HCC recurrence	
Notch2	Oncogene	Notch2 constitutive activation increases cell proliferation in vitro and in vivo	[51]
	Oncogene	Notch2 inhibition prevents tumor formation and decreases tumor burden	[39]
	Oncogene/stem-like properties	Knockout of Notch2 reduces cell growth and self-renewal	[52]
	Oncogene/stem-like properties	Notch2-positive HCCs display more immature cellular morphology	[53]
Notch3	Oncogene	Higher Notch3 levels correlate with more aggressive tumor traits and shorter survival	[54]
	Oncogene/drug resistance	Knockdown of Notch3 enhances sensitivity to doxorubicin in vitro	[55]
	Oncogene/drug resistance	Notch3 silencing increases the effect of brivanib in vivo	[56]
	Oncogene/drug resistance	Notch3 inhibition improves the efficacy of sorafenib in vitro and in vivo	[57]
Notch4	Oncogene/associated with HCC recurrence	Notch4 accumulation in HCC tissue is associated with high risk of recurrence	[58]
	Oncogene/EMT	Downregulation of Notch4 reduces cell migration and invasion in vitro, tumor growth in vivo	[59]

**Table 3 biomolecules-14-01581-t003:** Drugs targeting the Notch signaling pathway tested in preclinical and clinical settings.

Drug	Class	Observed Effects in Preclinical Studies	Development Phase	References
BMS-986115	γ-secretase inhibitors	Antitumor activity in xenograft models. GI toxicity, lymphoid depletion, and incomplete maturation of ovarian follicle	Phase I	[66]
LY900009		Inhibition of angiogenesis and tumor regression in Notch-dependent animal tumor models	Phase I	[67]
MK-0752		Reduction in BCSCs in breast tumorgraft models	Phase I, II, IV	[88]
LY3039478		Significant delayed tumor growth in several PDX models representing glioblastoma, iCCA, CCRCC, colon, gastric, lung, breast, and ovarian cancer. Inhibition of vessel formation in an in vitro angiogenesis model	Phase I, II	[68,69,70]
PF-03084014		Robust antitumor efficacy in Notch-driven models of T-ALL, CRC, breast, ovarian, and pancreatic cancer. Synergistic effects with docetaxel in TNBC models and with glucocorticoids in T-ALL xenograft mice	FDA approved for desmoid tumor treatment	[71,72,73]
RO4929097		Production of a less transformed, slower-growing phenotype in cell lines of colon, breast, melanoma, and pancreatic cancer. Antitumor effects on NSCLC xenograft models and absence of Notch-related toxicities	Phase I, II	[74]
AL101		Antineoplastic activity in xenograft models of T-ALL, colon, ovarian, pancreatic carcinoma, NSCLC, TNBC, HER2+ breast cancer, glioblastoma and neuroblastoma. No correlated toxicity was assessed in T-ALL1 and MDA-MB-468 xenografts	Phase I, II	[75]
MRK-560	γ-secretase modulator	Induction of cell cycle arrest in T-ALL cell lines and decrease in tumor burden in T-ALL patient-derived xenografts	Preclinical studies	[79]
INCB7839	ADAM inhibitors	Synergistic growth inhibition with lapatinib in breast cancer cells and in HER2+ breast cancer xenografts	Phase I, II	[89]
1H5		Inhibition of proliferation in colon cancer cell lines in vitro and enhanced effect on CRC xenografts in combination with the chemotherapeutic irinotecan without toxicity	Phase II	[81]
Brontictuzumab	mAb targeting Notch1	Decrease in proliferation, migration, and invasion of adenoid cystic carcinoma (ACC) cells and tumor burden in ACC xenografts. Inhibition of tumor growth and reduction in cancer stem cell frequency in a chemo-refractory BC xenograft model	Phase I	[90,91]
Tarextumab	mAb targeting Notch2/3	Antitumor efficacy in pancreatic, breast, ovarian, and small-cell lung cancer xenograft models	Phase I, II	[92]
PF-06650808	ADC targeting Notch3	Complete and durable tumor regression in TNBC cell line xenografts; tumor growth inhibition in LUSC PDX, NSCLC, and ovarian CLX models; significant antineoplastic effect in ovarian carboplatin-resistant tumors. Lack of general payload toxicity	Phase I	[93,94]
Rovalpituzumab tesirine	ADC targeting Dll3	Induction of durable tumor regression in SCLC and LCNEC PDX tumors. Reversible trilineage myelosuppression and mild kidney degeneration were observed in toxicology models	Phase I, II, III	[95]
MEDI0639	mAb targeting Dll4	A mouse DLL4 cross-reactive variant of MEDI0639 inhibits tumor growth and functional vessel formation in a number of human cancer xenograft models	Phase I	[96]
Demcizumab		Antitumor activity and decrease in cancer stem cell frequency in patient-derived xenograft models. Additionally, antiangiogenic effect and synergistic activity when combined with nab-paclitaxel in pancreatic cancer PDX models	Phase I, II	[97,98]
CB-103	Notch transcription complex inhibitor	Inhibition of mammosphere formation in combination with fulvestrant in ER+ cells and paclitaxel in TNBC cells. Significant tumor growth delay with paclitaxel in GSI-resistant TNBC models	Phase I, II	[86]

**Table 4 biomolecules-14-01581-t004:** Some of the crucial roles played by Notch signaling in the tumor microenvironment.

Cell Population	Function	References
Macrophages	Notch signaling regulates the polarization of macrophages into M1 and M2 types. Notch-activated macrophages negatively affect the antitumor activity of TILs	[103,107,111]
T cells	Notch signaling inhibits T-cell proliferation and activation by upregulation of PD-1 expression in CD4+ and CD8+ T cells. The Notch pathway also influences T-cell intratumoral infiltration	[108,109]
NK cells	Notch activation reduces intratumoral infiltration of NK cells	[108,109]
Tregs	Notch signaling regulates Treg frequency and activity within the TME	[109]
MDSCs	Notch signaling induces the expansion of MDSCs and promotes their recruitment within the TME	[108,109]

**Table 5 biomolecules-14-01581-t005:** Intrinsic roles of PD-1 and PD-L1 in cancer.

PD-1/PD-L1	Role/Involved Biological Process	Cancer Type	Putative or Observed Effect	References
PD-1	Oncogene	HCC	PD-1 blockade or knockdown inhibits tumor growth in vitro and in vivo	[142]
	Oncogene	TNBC	Knockdown of PD-1 reduces tumor growth and metastasis in vitro and in vivo	[143]
	Tumor suppressor	NSCLC	PD-1 blockade or silencing results in accelerated proliferation, in vitro and in vivo	[144,145]
	Tumor suppressor	Colon cancer	PD-1 blockade promotes cell proliferation	[144]
PD-L1	Oncogene/EMT	Lung cancer	PD-L1 knockdown reduces cell growth in vitro and in vivo, in human NSCLC models, whereas overexpression of PD-L1 promotes cell proliferation, migration, and invasion	[147]
	Oncogene/ EMT/drug resistance	Gastric cancer	PD-L1 knockdown decreases cell proliferation, migration, and invasion in vitro and in vivo. It also increases sensitivity to CIK therapy in vitro	[149]
	Oncogene/ EMT/drug resistance	Breast cancer	PD-L1 knockdown halts the expression of multidrug resistance 1/P-glycoprotein. In TNBC tumor models, PD-L1 depletion attenuates migration and lung metastasis	[148,151]
	Tumor suppressor	Cholangiocarcinoma	Knockdown of PD-L1 favors cell cycle arrest and tumor CSC-like phenotypes	[117]

**Table 6 biomolecules-14-01581-t006:** Roles of the PD-1/PD-L1 signaling pathway in immune cells.

Cell Population	Function	References
T cells	Regulation of T-cell activation; upon binding, PD-1 and PD-L1 on T cells transduce an intracellular suppressive signal	[122]
CD8+ central memory T cells	PD-1 signaling negatively regulates CD8+ central memory T-cell differentiation into effector cells	[155]
B cells	Regulation of B cell activation; PD-1/PD-L1 axis hinders BCR-mediated activation and antibody secretion	[156]
Natural killer cells	Regulation of NK cell effector functions; PD-1 hampers the degranulation of NK cells	[153,157]
Neutrophils	PD-L1 signaling inhibits neutrophil antitumor cytotoxicity	[158]
Dendritic cells	PD-L1 on DCs negatively regulates T-cell responses and protects DCs from T-cell cytotoxicity	[160]
Tregs	PD-1/PD-L1 axis controls CD4+ T-cell transformation into Tregs and enhances their activity	[153]
TAMs	PD-1/PD-L1 signaling induces M2-type macrophage polarization. PD-1 negatively regulates TAM effector functions against tumor cells	[161]

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
