# Peer review of "Notch Signaling and PD-1/PD-L1 Interaction in Hepatocellular Carcinoma: Potentialities of Combined Therapies"

_biomolecules, 2024, doi:10.3390/biom14121581_

Round 1
Reviewer 1 Report
Comments and Suggestions for Authors
Here are some suggestions for improving the article as part of the peer review process:
Major comments
1. Clarify the Mechanisms of Notch and PD-1/PD-L1 Interplay
The article discusses the crosstalk between Notch signaling and the PD-1/PD-L1 axis but could benefit from a clearer explanation of the mechanisms involved. Adding more detail on how Notch signaling directly influences PD-L1 expression and vice versa, especially in HCC, would strengthen the discussion. This could include visual aids like pathway diagrams to clarify these interactions.
2. Expand on Clinical Implications and Potential Challenges
While the article advocates for combining Notch inhibitors with ICIs for HCC treatment, it should address potential clinical challenges, such as:
Toxicity and Side Effects: Discuss any known or anticipated side effects of Notch inhibitors and ICIs, especially when used in combination, and how these might be mitigated.
Patient Selection: The article mentions using exosomes as biomarkers but could go further by discussing specific criteria or protocols for selecting patients who would benefit most from this combination therapy.
Resistance Mechanisms: Explore how resistance to either Notch inhibitors or ICIs could affect the efficacy of the combined treatment and potential strategies to overcome such resistance.
3. Discuss the Potential for Non-Invasive Biomarkers in Greater Detail
The use of exosomes as non-invasive biomarkers is a compelling aspect of the article. Expanding on this topic would enhance its relevance. Suggested additions could include:
l Specific molecular markers within exosomes that indicate Notch activation status.
l The current state of research on using exosomes in HCC and how reliable or feasible this approach is in clinical settings.
l Any technical challenges related to isolating and analyzing exosomes, as well as proposed solutions.
4. Include Data from Preclinical and Clinical Studies
If available, incorporating findings from preclinical studies or early-phase clinical trials that test the combination of Notch inhibitors and ICIs in HCC could make the article more persuasive. Even if limited, reference to ongoing trials or recently published studies on similar strategies in other cancers (where applicable) would provide valuable context.
5. Address Notch's Dual Role in Cancer More Clearly
The article acknowledges that Notch can act as both an oncogene and a tumor suppressor depending on the context. Expanding on this dual role in the context of HCC would add depth. For instance:
l Provide specific examples of when Notch signaling promotes tumor progression in HCC and when it suppresses it.
l Discuss how the role of Notch signaling in the tumor microenvironment affects this duality, particularly concerning immune-cell interaction.
6. Provide a More Detailed Future Research Roadmap
In addition to suggesting future research directions, the article could lay out a clearer roadmap for moving from theory to clinical practice. This could include:
l Proposed steps for validating exosome-based biomarkers.
l Detailed research phases for studying Notch and PD-1/PD-L1 combination therapies, from cell-based assays and animal models to clinical trials.
l Specific methodologies or experimental designs that could help elucidate the mechanisms of action and optimize treatment protocols.
7. Enhance Visual Aids and Figures (Section 9. The Crosstalk between the Notch Pathway and PD-1/PD-L1 Axis)
Visual representations of complex pathways like Notch signaling and the PD-1/PD-L1 axis would make the article more accessible. Additionally, schematics of the proposed combination therapy mechanism, biomarkers, and potential therapeutic outcomes could strengthen the article’s impact and clarify the concepts for readers.
8. Conclude with Specific Clinical and Translational Recommendations (Section 10. Conclusions page 15 lines 684-723)
The conclusion could be more actionable by offering specific recommendations for clinicians and researchers. For instance:
l Guidelines for incorporating Notch signaling assessment in HCC treatment planning.
l Suggestions for how oncologists might interpret Notch and PD-1/PD-L1 activity levels in HCC when deciding on therapy combinations.
l Potential strategies for implementing these findings in early clinical trials or pilot studies.
These suggestions aim to enhance the clarity, depth, and clinical relevance of the article. Including more specific mechanisms, expanding on clinical challenges, and providing a detailed research roadmap would strengthen the article's impact and utility for the oncology research community.
Minor comments:
9. Figure 1. The text in this figure is a bit blurry. May recreate it for better clarity.
10. Please correct PD-1/PDL-1 to PD-1/PD-L1 on lines 367, 453, and 556.
Comments on the Quality of English LanguageEnglish could be improved
Reviewer 2 Report
Comments and Suggestions for Authors
The paper is nicely written.
My only concern is can authors please summarize the NOTCH targeted drugs and their effect on tumor growth in tabular form?
Also, please discuss the clinical trials involved targeting NOTCH to treat various cancers.
Reviewer 3 Report
Comments and Suggestions for Authors
This review reports on the interplay between Notch signaling and the PD-1/PD-L1 axis in HCC
My comments:
The review is well-written and comprehensive
Provide us with a table of the studies described for: Notch signaling in epithelial cells; in the liver, in immune cells (chapters 3-5) and also for PD-1/PD-L1
Conclusions: is not very precise. Sharpen/shorten the text and highlight your conclusions
Round 2
Reviewer 1 Report
Comments and Suggestions for Authors
The currently revised article can be accepted for publication